# Rapid and Differential Evolution of the Venom Composition of a Parasitoid Wasp Depending on the Host Strain

**DOI:** 10.3390/toxins11110629

**Published:** 2019-10-29

**Authors:** Fanny Cavigliasso, Hugo Mathé-Hubert, Laurent Kremmer, Christian Rebuf, Jean-Luc Gatti, Thibaut Malausa, Dominique Colinet, Marylène Poirié

**Affiliations:** Université Côte d’Azur, INRA, CNRS, ISA, 06 903 Sophia Antipolis, France; fanny.cavigliasso@gmail.com (F.C.); Hugomh@gmx.fr (H.M.-H.); Laurent.Kremmer@inra.fr (L.K.); Christian.rebuf@inra.fr (C.R.); Jean-luc.gatti@inra.fr (J.-L.G.); thibaut.malausa@inra.fr (T.M.); dominique.colinet@inra.fr (D.C.)

**Keywords:** parasitoid wasp, venom composition, experimental evolution, 1D protein electrophoresis, host specificity, *Leptopilina boulardi*, Drosophila melanogaster

## Abstract

Parasitoid wasps rely primarily on venom to suppress the immune response and regulate the physiology of their host. Intraspecific variability of venom protein composition has been documented in some species, but its evolutionary potential is poorly understood. We performed an experimental evolution initiated with the crosses of two lines of *Leptopilina boulardi* of different venom composition to generate variability and create new combinations of venom factors. The offspring were maintained for 10 generations on two strains of *Drosophila melanogaster* differing in resistance/susceptibility to the parental parasitoid lines. The venom composition of individuals was characterized by a semi-automatic analysis of 1D SDS-PAGE electrophoresis protein profiles whose accuracy was checked by Western blot analysis of well-characterized venom proteins. Results made evident a rapid and differential evolution of the venom composition on both hosts and showed that the proteins beneficial on one host can be costly on the other. Overall, we demonstrated the capacity of rapid evolution of the venom composition in parasitoid wasps, important regulators of arthropod populations, suggesting a potential for adaptation to new hosts. Our approach also proved relevant in identifying, among the diversity of venom proteins, those possibly involved in parasitism success and whose role deserves to be deepened.

## 1. Introduction

The question of the adaptive evolution of venom composition has been mainly studied in predatory venomous animals, with evidence for natural selection generally driven by diet (e.g., [1,2,3]) but also by environmental conditions [4]. This question remainshowever largely unaddressed for parasitoid wasps—insects that develop at the expense of their arthropod host, leading to its death—although they rely primarily on the injection of venom during oviposition to overcome the immune defenses of the host (reviewed in [5,6,7]) and optimize the development of their offspring [8,9]. Parasitoid wasps act as important regulators of insect species communities in the field and are thus widely used as biological control agents. Understanding whether rapid changes can occur in the composition of their venom is therefore a key issue in assessing their ability to adapt to new hosts, whether in the wild or under rearing conditions.

Variations in venom composition, a prerequisite for venom evolution, have been documented both between parasitoid taxa and, more recently, intraspecifically, e.g., in *Leptopilina boulardi* (Hymenoptera: Figitidae; [10,11]) and the *Psyttalia* genus (Hymenoptera: Braconidae; [12]). In *Leptopilina*, the closely related species *L. heterotoma* and *L. boulardi* have no abundant venom proteins in common, while the two well-characterized *L. boulardi* laboratory isofemale lines ISm and ISy share roughly no more than 50% of the abundant venom proteins [10]. The variation of venom composition between ISm and ISy is probably mainly due to a quantitative differential expression of alleles in the two *L. boulardi* lines, as demonstrated for LbGAP [13]. LbGAP is a RhoGAP domain-containing protein seemingly required for parasitism success on the resistant strain of *D. melanogaster* (see below) through targeting of the host immune cells [13,14,15,16,17]. This protein is much more abundant in the ISm venom than LbGAPy in the ISy venom due to a variation in the *cis*-regulation of the expression of the two alleles [13]. The variation between ISm and ISy venoms can however be also qualitative as exemplified by LbSPN, a serine protease inhibitor of the serpin superfamily. LbSPNm (ISm) and LbSPNy (ISy) are found in considerable amounts in both lines, but, although encoded by alleles of the same gene, they differ in their molecular weight and likely in their targets due to large differences in the active site [10]. LbSPNy was demonstrated to target the phenoloxidase cascade, a key component of the immune system, in *Drosophila yakuba* host larvae [18]. A role for LbSPNm in parasitism success is yet to be made evident [10]. The venom composition was also shown to vary between *L. heterotoma* and *L. boulardi* individuals from natural populations [11,19], suggesting a high evolvability. Accordingly, geographical variations in venom composition were observed and demonstrated to be partly due to some local adaptation in the case of *L. boulardi* populations [19].

The *L. boulardi* isofemale lines ISm and ISy not only differ in their venom composition, but also in their virulence properties. ISm is highly virulent on *D. melanogaster,* but eggs are always encapsulated by the tropical species *D. yakuba*. In contrast, ISy can succeed on both host species but depending on their resistant/susceptible phenotype [20,21]. Genetic approaches and transfer of chromosomes between a *D. melanogaster* resistant and a susceptible line to ISy parasitoids have allowed identifying one major diallelic resistance gene, *Rlb*, located on the right arm of chromosome 2 [22,23,24]). The two final strains obtained, one resistant (R) and one susceptible (S) to ISy (Gif n°1088 and 1089), used in this work, differ mainly in the chromosome 2 on which *Rlb* is located (2R) [25]. Candidates loci for this gene were then later identified [25,26].

Here, we report data from an experimental evolution approach designed to determine whether the venom composition of *L. boulardi* can evolve rapidly according to the host strain. Hybrids with new venom allelic compositions were obtained by crossing ISm and ISy individuals and then maintaining the offspring on the (R) vs. (S) *D. melanogaster* strains for 10 generations. The venom composition of individual wasps was then analyzed in F_2_, F_6_ and F_10_ using two different approaches. The recently developed global approach is based on one dimensional SDS-polyacrylamide gel electrophoresis (1D SDS-PAGE) coupled with further statistical analysis of the intensity of the protein bands. It allows analyzing the quantitative variation for large sets of proteins directly from individuals [19,27]. This approach identified a number of venom bands whose intensity has changed on a given host strain. The most abundant proteins in these bands were then identified using previous “omics” data obtained from *L. boulardi* ISm and ISy venom [10]. To check the accuracy of the global approach, we used a specific approach based on Western blot analysis designed to study the evolution of the amount of three well-characterized venom proteins, LbGAP, LbSPN (described above) and LbGAP2, one of the eight other RhoGAP domain-containing proteins found in the *L. boulardi* venom [10]. Overall, we observed a rapid and differential evolution of the venom composition on the two host strains, and proteins whose quantity in venom has probably changed under selection were identified. These are candidates for further functional studies to assess their possible role in parasitism success.

## 2. Results

### 2.1. Experimental Evolution Protocol

To create variability in the venom composition on which selection can take place, we performed an experimental evolution using F_1_ hybrids between the laboratory isofemale lines ISm and ISy. F_1_ hybrids were produced by 12 independent couples (ISm female × ISy male; see Appendix A for a synthesis of interaction outcomes between *D. melanogaster* and *L. boulardi* strains). Within the progeny of each of the 12 couples, each corresponding to one replicate, we created two groups by randomly picking 10 F_1_ females and 5 F_1_ males that were given either the (S) or the (R) host strain larvae as hosts. Each of the following generations, up to F_10_, was created in the same way and consistently maintained on the same host strain (see Material and Methods and Appendix A). Of the 12 replicates created by the initial couples, only eight could be kept on both hosts until the F_10_. The 16 groups (eight on each host strain) are hereafter referred as “populations”. Specific and global analyses of venom composition, respectively based on image analysis of Western blots and 1D SDS-PAGE (Appendix A), were made on a total of 474 females from three generations: F_2_ (1st generation that we considered under selection on the 2 hosts, see Material and Methods and Appendix A), F_6_ (intermediate generation), and F_10_. Ten females were used for each of the eight replicates on the two host strains and the three generations, except for one F_2_ replicate on the (S) host for which only four individuals were available.

### 2.2. Host Specific Evolution of Venom Composition Evidenced by a Global Approach

The venom protein content of each female tested was separated on 1D SDS-PAGE and the resulting protein profiles of individual wasps were analyzed using a semi-automated method [28]. We could identify 34 reference bands whose intensities represent the variables describing the composition of the venom (Figure 1). Analysis of band intensity variation with a permutational MANOVA (Table 1) revealed a significant effect of the “host” (*p* < 0.001), the “generation” (*p* = 0.02), and the “host × generation” interaction (*p* = 0.001). To assess whether this interaction resulted from a different selection on both hosts or selection on a single host only, MANOVAs were performed separately for the (R) and (S) host strains, showing a significant effect of the generation (Table 1; *p* = 0.001 and *p* = 0.002 for the (R) and (S) strains, respectively). However, although highly significant, the combined effects of the “host”, “generation”, and the interaction of the two only accounted for a small part of the variance of the venom composition (R² < 0.04; Table 1) compared to that explained by the replicate population (0.14 < R² < 0.21).

To further characterize the evolution of venom composition as a function of the host, we performed a linear discriminant analysis (LDA). Although the six groups (two hosts and three generations) widely overlapped—in keeping with the small part of the variance explained by the MANOVA—the LDA was significant (*p* < 0.001). The first and second of the five discriminant axes identified were the only ones to be biologically meaningful: They discriminated the three analyzed generations (F_2_, F_6_ and F_10_) for the replicates evolved on the (S) or (R) strains (Figure 2, Appendix A) and could be interpreted as “venom evolution on the (S) host” and “venom evolution on the (R) host”, respectively (Figure 2).

### 2.3. Identification of Protein Bands Whose Intensity Has Changed by Selection on Different Host Strains

We first identified the selected protein bands based on the correlation of the intensity of the bands to the two LDA axes that describe the evolution on the two host strains. Based on Spearman rank correlations, most of the protein bands (27 out of 34) were found to be significantly correlated to at least one of the two LDA axes (Table 2, Appendix A). A Spearman rank correlation test also revealed that the position of the bands on the two axes were negatively correlated (Spearman correlation: −0.72, *p* < 1.10^−3^), indicating that most of the evolved bands were selected on both hosts, but in opposite directions. Overall, in terms of intensity, bands that evolved to a higher intensity on the (R) host evolved to a lower one on the (S) host and vice versa.

Some of the 27 protein bands could have been selected only indirectly because of their correlation with other directly selected bands, either due to linkage disequilibrium or to their proximity to another band on the gel (see [27]). To distinguish indirectly selected bands from those responding directly to the selection pressure, we used partial correlations as reported for an artificial indirect selection [28]. Specifically, the 34 protein bands were first grouped into seven clusters of bands with at least 40% correlation (threshold arbitrary chosen to be conservative) (Figure 3A). Then, we performed a partial correlation analysis within the five clusters containing at least two bands significantly correlated to the same axis. This made it possible to determine which bands in a cluster remained correlated to one or both discriminant axes independently of the other bands in the cluster (Table 2, Appendix A). At the end, 15 out of the 34 protein bands remained significantly correlated to one of the two LDA axes i.e., showed a change in intensity on at least one of the two host strains (Table 2, Appendix A; Figure 3B). Among these, (i) 4 and 3 bands were positively and negatively selected on the (S) host respectively, (ii) 3 bands were negatively selected on the (R) host, (iii) 3 bands were both positively selected on the (S) host and negatively on the (R) host, (iv) 1 band was negatively selected on the (S) host and positively on the (R) host, and (v) 1 band was positively selected on both hosts (Table 2, Appendix A, Figure 3B).

### 2.4. Putative Identification of Venom Proteins Whose Amount Evolved in Response to the Host Strain

In a next step, we sought to identify the proteins contained in the bands under selection by matching the 15 selected bands identified in the global approach (Figure 1) with the bands on the 1D electrophoresis gels used for *L. boulardi* venom proteomics [10]. Since one band can contain different proteins, these proteins can be responsible for the level of intensity of bands migrated at the same position. However, since the initial composition of the venom resulted from hybridization of ISm and ISy, the proteins responsible for high band intensity were mainly expected to be the most abundant proteins in the corresponding band of ISm or ISy venom. Based on this reasoning, the number of peptides matches from mass spectrometry was used to classify the proteins in the bands as abundant or not (Table 3). We could identify at least one abundant protein for 10 out of the 15 bands detected as directly selected by the global approach (Table 3). In these bands, low abundant proteins are unlikely to have driven the observed changes in the whole band intensity.

For 4 of these 10 bands, the most abundant proteins had no similarity with known proteins and their function is therefore unknown (Table 3), thus underlining the relevance of the global approach to identify candidate venom proteins on which future studies could focus. In band #11, selected on the (S) strain, a Sushi/SCR/CCP domain-containing protein (SCR: Short Consensus Repeat; CCP: Complement Control Protein (CCP); also identified in band #5 but not as the most abundant in that band) was found in the same amount as a protein with unknown function). For the five other bands, the most abundant proteins with a predicted function (Table 3) are (i) LbSPNy (band #20), counter-selected on the (R) host, (ii) LbGAP (band #24) counter-selected on the (S) host, (iii) LbGAPy4 (band #26), another RhoGAP positively selected on the (R) host and counter-selected on the (S) host, and (iv) LbGAP2 (the most abundant protein in bands #28 and #29), counter-selected on the (R) host in both bands and selected on the (S) host in band 29 (Table 3).

### 2.5. Trends of Venom Band Evolution

To further explore the trend of venom evolution on the two hosts, we evaluated the proximity of each of the selected protein bands to the ISm or ISy bands, by comparing their respective intensities. Each band was thus assigned a value between 0 and 1, 1 meaning the band is present in ISm and absent in ISy and 0 the opposite (see material and methods, Table 2 and Appendix A). This analysis revealed that bands positively selected on the (S) host (6 out of 8) and bands counter-selected on the (R) host (4 out of 6) were more intense in the ISm venom profile than in the ISy one (Table 2).

### 2.6. Host Specific Evolution of Venom Composition Evidenced by a Specific Approach

To assess the accuracy of the global approach, we performed Western blot analyses to specifically track the evolution of the amounts of three well-characterized venom proteins found as the most abundant in one or two of the selected protein bands (see above): the serpin LbSPN and the RhoGAPs LbGAP and LbGAP2 (Figure 4A).

The variation for LbSPN has a simple genetic determinism, the two codominant alleles of the *lbspn* gene encoding proteins of different molecular weight (LbSPNy 54 kDa in ISy and LbSPNm 45 kDa in ISm). We therefore used a simulation approach to build the null distribution of a summary statistics assessing the distance between the observed and expected frequencies under the assumptions of neutrality (H0) and panmixia (i.e., no population structure and random mating; Appendix A). Results showed that the frequency of the *lbspn_y_* allele was significantly lower than expected on both host strains (*p* = 0.047 and *p* = 0.045, for the (R) and (S) strains, respectively; Figure 4B, Appendix A). This was partly consistent with data from the global analysis showing that band #20 (in which LbSPNy is the most abundant protein) was counter-selected on strain (S). However, band #20 was not counter-selected on the strain (R) and band #21 whose most abundant protein is LbSPNm was not selected on either host.

The gene encoding LbGAP also has two alleles but *lbgap*, overexpressed in the venom gland of ISm, is dominant over *lbgap_y_* (ISy). Accordingly, LbGAP is detected as a strong signal in the venom of ISm, F_1_, or F_2_ females (homozygous or heterozygous for *lbgap*), but it is barely detectable in ISy individuals. Using a similar approach as for LbSPN—except that a dominant allele was simulated—LbGAP was shown to be selected on the (R) host (*p* = 7 × 10^−3^; Figure 4C, Appendix A) with the *lbgap* allele fixed in five out of the eight replicates. F_6_ and F_10_ showed a similar proportion of individuals with a high LbGAP quantity, suggesting early selection in the experimental evolution (Figure 4C). Band #24 in which LbGAP is the most abundant protein did not appear as selected on the (R) host. Indeed, the observed positive correlation with the corresponding axis was only significant before the partial correlation analysis (Table 2).

Since the genetic determinism of LbGAP2 was unknown, we analyzed it as a continuous variable using a linear mixed model and a Tukey test for multiple comparisons. The amount of LbGAP2 was shown to decrease between F_2_ and F_6_ on the (R) host (Tukey post hoc test, *p* < 0.05) while it did not change on the (S) host (Figure 4D; Appendix A). Results thus suggest that LbGAP2 was counter-selected on the resistant host early in the experimental evolution. This is consistent with the global analysis data for bands #28 and #29 in which LbGAP2 is the most abundant protein. In contrast to the specific analysis, however, band #29 appeared selected on the (S) host suggesting the selection of one or more other venom proteins in the band.

## 3. Discussion

The venom of parasitoids is a key success factor. It contains a wide variety of components, mainly proteins, that ensure parasitism success by counteracting the immune response of the host, interfering with its development or manipulating its physiology (reviewed in [5,6,7]). Parasitoids may be confronted with changes in their environment and have to adapt to new hosts. Thus, we tested (i) whether venom composition could evolve differently on well-characterized strains of the same host species, showing a contrasted phenotype of resistance to two lines of *L. boulardi*, ISm and ISy, and (ii) whether this evolution could take place in a few generations. We performed an experimental evolution initiated by crossing these two parasitoid lines whose composition of venom is very different [10]. Resulting individuals were then separated in independent populations reared either on the resistant (R) host strain or the susceptible (S) host strain. From F_2_, which harbor new random combinations of ISm and ISy venom factors, individuals were maintained consistently on these same host strains until the F_10_ generation. Since the two ISm and ISy isofemale lines have passed many generations in the laboratory, the initial standing variation on which natural selection may have operated during the experimental evolution most likely results primarily from the variation between ISm and ISy.

The venom composition of the F_2_, F_6_ and F_10_ females was characterized for each individual by a “without a priori” approach based on the variations in intensity of reference protein bands on 1D gels, mainly reflecting the quantitative variations of the proteins (more details in [27]). The accuracy of the results was checked by a specific approach based on Western blot analysis of three well-characterized venom proteins. Both approaches proved congruent and revealed a rapid differential evolution on the two host strains (see below). Indeed, although the specific approach was more powerful in detecting changes, the well-characterized proteins and the protein bands in which they were abundant were generally detected as evolving in the same way in both analyses. A significant effect of genetic drift on venom composition was expected due to the small size of the population formed in the experiment. We could however identify changes in venom composition common to most replicated populations. Since genetic drift is a purely random phenomenon that has affected each population differently, these changes most probably result from selection-driven evolution. Results therefore suggest that venom selection was strong enough not to be masked by genetic drift.

### 3.1. The Composition of Venom Evolves Rapidly Under Selection on Host Strains

The evolution of the composition of the venom was detectable after only four generations of selection, which is very fast, and this despite strong variation between replicates and individuals. In generation 2, the first considered to be subject to selection, no differentiation in venom composition was observed between individuals maintained on the Drosophila (R) and (S) hosts. There was therefore no detectable plasticity effects related to the host strain on venom composition. Consequently, the divergent evolutionary trajectories observed afterwards were most likely genetically determined.

Overall, we detected changes in the intensity of up to 15 protein bands (out of 34 reference bands). It is unlikely that all these evolving bands were selected in response to the immune defense of the host. The (R) and (S) strains not only differ in their genotype at the *Rlb* resistance gene, but also in a large part of chromosome 2 on which this gene is located [22,23,24], suggesting possible selection pressures other than those associated with the [Rlb] phenotype. In addition, the (R) and (S) strains differ in the timing of the peak of the number of circulating hemocytes observed after parasitism [29] and the best candidate gene identified for *Rlb* is part of a key signaling pathway in cell proliferation and differentiation [26]. *Rlb* could thus have pleiotropic effects on host physiology, the selection for optimal parasitoid development on each host strain leading to differential selection of venom on the both hosts.

### 3.2. A Majority of Bands Whose Intensity Changed Are of ISm Origin

A majority of the bands whose intensity changed (9 out of 15) were originally more intense in the venom of ISm suggesting their ISm origin. Moreover, although both ISm and ISy succeed in developing in the (S) strain, 6 of the 8 bands selected on the (S) host were of ISm origin. As ISm is more specialized on *D. melanogaster*, its venom factors could be more effective on this host species than those of ISy, accounting for the largest number of selected bands of ISm origin in the experiment. ISy parasitoids originate from a tropical region in which the number of suitable *Drosophila* host species is higher than in the Mediterranean region (ISm origin). This would explain their lesser success on a given host (dependent on the host resistance) but on a larger number of species [20,30,31,32]. ISy venom proteins should therefore be less effective on *D. melanogaster* than those of ISm.

Surprisingly, only one band was positively selected on the (R) strain, suggesting that the higher virulence of ISm on the (R) host would depend on a limited number of factors in its venom. However, two ISm proteins, LbGAP and LbSPNm, were identified as being selected on the (R) host in the specific analysis. The global analysis might thus have underestimated the number of ISm venom bands that were selected on the (R) strain. Another striking observation is that most of the ISm bands whose intensity changed on the (R) strain were counter-selected, suggesting that they contain useless and perhaps even costly proteins on this host. This is surprising since the frequencies of resistance/susceptibility to the ISy line are roughly similar in *D. melanogaster* populations worldwide, especially in the Mediterranean area from which the ISm line originates [20]. A venom composition effective on both susceptible and resistant hosts should therefore have been selected in ISm-like parasitoids, with less counter-selection than observed.

### 3.3. Venom Components Can Be Costly

The existence of a cost for a number of venom proteins is supported by the negative relationship between the correlations of four of the bands whose intensity changed (#3, 5, 26 and 29) with the axes 1 and 2. This suggests that these bands were positively selected on one host strain and counter-selected on the other. Counter-selection was also observed for two of the three proteins tested in the specific analysis (see the next paragraph). This is also in agreement with the reported decay of virulence of the ISy line towards *D. yakuba* when reared on *D. melanogaster* [31]. The cost may be due to the energy consumption required to produce venom proteins, most of which are abundant, as observed for some venomous taxa [33], leading to trade-offs between the production of different proteins in the body, or between venom proteins and other biological functions. The compromises could also concern the venom factors themselves, in relation to their respective activity or function. Some venom-related factors may also directly impact the parasitoid fitness due to side effects in the venom apparatus or in the host. Finally, since ISm and ISy strongly differ in their venom composition, the experimental design led to creation of new combinations of venom factors—with possible epistasis or redundancy effects—and the simultaneous presence of some of them could increase or decrease the parasitoid fitness on the given host.

### 3.4. Proteins Whose Quantity Potentially Evolved

The protein content of the bands whose intensity changed was tentatively identified by manually matching them with the 1D electrophoresis gels previously used to characterize the venom proteins of *L. boulardi* ISm and ISy lines [10]. Of the most abundant proteins identified, most likely responsible for changes in band intensity, a majority do not have a predicted function. Proteins for which no putative function can be predicted from similarity searches are commonly found in the venom of parasitoids but rarely studied. Our unprecedented prior-less global approach is therefore particularly relevant in identifying among the diversity of venom proteins, those for which the role in parasitism is worth studying.

Other identified venom proteins whose amount had potentially changed include LbGAP, LbGAP2and LbGAPy4, members of the RhoGAP family. According to the specific approach, LbGAP was positively selected on the (R) host, in agreement with its apparent involvement in parasitism success on this strain [13]. LbGAP was indeed demonstrated to be transported by venosomes (venom vesicles) and thus targeted and internalized inside the host lamellocytes [17]. The morphology of these immune cells is then altered probably through the inactivation of Rac1 and Rac2, demonstrated to be targeted by LbGAP in vitro and both required for the host immune response [14,15,16,34,35] Band #24, in which LbGAP is the most abundant, was however not detected as positively selected on the (R) strain possibly due to our choice of being conservative in the global approach (use of a partial correlation approach followed by a Bonferroni correction). Indeed, band #24, detected as positively correlated to the axis corresponding to the (R) strain before the partial correlation, belongs to cluster #2 for which the partial correlation analysis failed to identify the bands directly correlated to this axis. In addition, the dominance of the LbGAP ISm allele (*lbgap*) combined with its high frequency in initial F_1_ individuals may have prevented the detection of positive selection in the global analysis. LbGAP2 is a venom RhoGAP protein mutated on the catalytic site as all other RhoGAPs identified in venom except LbGAP, making them likely inactive as RhoGAPs [10]. Although its possible role in parasitism success is yet unknown, it has recently been shown to be transported by venosomes and targeted and internalized inside host lamellocytes as LbGAP [17]. LbGAP2 was identified as the most abundant protein in band #28, counter-selected on the (R) host, and band #29, positively selected on the (S) strain and counter-selected on the (R). The specific analysis identified only the counter-selection on the (R) host, thus suggesting that the increase in intensity of band #29 on the (S) strain is due to one or more of the other abundant proteins in that band. Finally, LbGAPy4, another mutated RhoGAP found in ISy venom, was detected as the most abundant protein in band #26, positively selected on the (R) strain, thus suggesting a role for this protein in parasitism success as well.

Among the identified proteins whose amount had potentially changed is LbSPNy found as counter-selected on the (R) strain in both global and specific analyses. LbSPNy is involved in suppressing the immune response of another host species, *D. yakuba*, through the targeting of the phenoloxidase cascade which plays a crucial role in defense against parasitoids [18]. LbSPNy is found in the venom of ISy and does not seem to be involved in parasitism success against *D. melanogaster*. It is therefore probably costly and counter-selected on this species.

Most of the rapid and differential evolution of the venom protein composition we observed results from quantitative differences in venom proteins previously observed between the two lines of *L. boulardi* [10]. This brings us to the question of the molecular origin of this variability. Although several mechanisms may be involved, quantitative differences may largely rely on changes in *cis*-regulation of gene expression, as previously demonstrated for LbGAP between ISm and ISy [13]. Accordingly, it was recently demonstrated that the process of recruitment and loss of genes encoding venom proteins in four related parasitoid wasp species is due to changes in *cis*-regulation of their expression [36]. Moreover, differences were recently reported in the expression levels of genes encoding putative venom proteins of a parasitoid wasp, in response to the symbiont-associated resistance of the host [37]. Such changes would favor a rapid divergence in venom protein composition even between closely related parasitoids and individuals of the same species. The study of changes in *cis*-regulatory sequences will largely benefit from ongoing genome sequencing and annotation projects.

Overall, this work highlights the capacity for rapid and partially adaptive evolution of the venom composition of parasitoid wasps, important regulators of arthropod populations in ecosystems and biological control auxiliaries. In a context of climate change and of the increase in the number of invasive arthropod species, this is an important asset, making it possible to adapt to these new hosts. Such a capacity for rapid adaptation may, however, cause issues when rearing parasitoid wasps on alternative hosts for biological control since a change in venom composition may impact their ability to successfully parasitize the target host in the field.

Our study relied on laboratory lines to investigate the evolution of venom composition on two strains of the same host species. The results of this laboratory experiment are an invitation to study the eco-evolutionary dynamics of this composition in the field. We have recently shown that it is possible to distinguish neighboring natural populations of *L. boulardi* based on the composition of their venom [19]. This pattern seems to involve some local adaptation either (i) to differences in local abiotic conditions, or (ii) to intraspecific variations within a given host species, or (iii) to variations in the community of host species. The fact that the venom can adapt rapidly and differently depending on the host strain suggests that it could also adapt differently to different host species. This specialization of the venom composition will depend on the level of specificity of the venom proteins, as well as on the cost associated with their presence if the species on which they are effective is no longer present. Thus, variations in the community of host species in the field may leave visible traces on the venom composition of the associated parasitoids. We have addressed this issue of the specialization of the venom composition on different host species in a new experimental evolution [38].

Finally, this type of study allows identifying selected venom proteins in a specific condition and deciding on which to focus, even if they have no predicted function. It can also highlight particularly effective protein associations. As a result, markers of parasitic success could be defined allowing a quality control of the species used in biological control.

## 4. Materials and Methods

### 4.1. Biological Material

The origin of the isofemale lines of *L. boulardi* ISy (Gif stock number 486) and ISm (Gif stock number 431) isofemale lines has been previously described [39]. Briefly, the ISy and ISm founding females were collected in Brazzaville (Congo) and Nasrallah (Tunisia), respectively. Both lines were reared on a susceptible *D. melanogaster* strain (Gif stock number 1333) at 25 °C. After emergence, wasps were kept at 20 °C on agar medium with honey.

The *D. melanogaster* (R) strain (Gif number 1088) and (S) strain (Gif number 1089) are respectively resistant and susceptible to *L. boulardi* ISy. They were obtained from isofemale lines of a population of Brazzaville, Congo [22] through subsequent genetic approaches [40], and they only differ by a large part of the chromosome 2 that contains a major resistance gene to ISy parasitoids.

### 4.2. Experimental Evolution Protocol

We used a classical experimental evolution protocol [41] to evaluate the potential for evolution of the venom composition of *L. boulardi*. The variability on which the selection could occur was generated by crossing the two ISm and ISy *L. boulardi* lines. These hybrids were then raised on the susceptible (S) or on the resistant (R) strain of *D. melanogaster*. ISm is virulent on both (R) and (S) strains, whereas ISy only succeeds on the (S) strain (Appendix A). Each of the eight analyzed replicates was obtained from the F_1_ offspring of one individual cross (ISm female × ISy male) by creating two groups of 10 females and 5 males chosen at random, which were then separately maintained on resistant or susceptible *D. melanogaster* flies. The following generations, also created with 10 females and 5 males chosen at random, were maintained on the same host up to F_10_ (Appendix A). Replicates were organized in randomized positions to ensure that any uncontrolled environmental effects were the same on both treatments (R) and (S). This also ensures that any changes of such uncontrolled environmental effects during the experiment would affect both treatments in the same way. The individual venom composition was analyzed for three different generations: (F_2_) females produced by (ISm x ISy) hybrids. F_2_ is considered here as the first generation under selection since it is the first that displays a variety of combinations of ISm and ISy alleles that will go through the filter of the selection on the two hosts. (F_6_) females from an intermediate generation of selection, (F_10_) females from the last generation of selection. Since the amount of venom may vary between females according to the number of eggs previously laid, only females never allowed to parasitize were used for analysis of the venom. The remaining variation was considered as described below.

### 4.3. Venom Analysis and Data Acquisition

*L. boulardi* venom reservoirs were dissected individually in 15 µL of insect Ringer solution supplemented with protease inhibitors cocktail (PI; Roche), mixed with an equivalent volume of Laemmli reducing buffer and heated (95 °C, 10 min). These individual venom protein samples were then split in two. One half was used for the global analysis and the other for the specific analysis (see below). Proteins from each of these venom samples from an individual female wasp were separated by 1D SDS-PAGE using commercial gels to increase reproducibility (Any-kD Mini-PROTEAN^®^ TGX™; Bio-Rad, Marnes-la-Coquette, France). In addition, samples of venom of ISm and ISy lines, equivalent to half a female reservoir, were loaded on each gel to serve as a reference (Figure 1A). These reference samples were obtained from a pool of venom from 10 individuals collected in 300 µL. The first lane of each gel was dedicated to the molecular weight marker (5 µL, Spectra™ Multicolor Broad Range Protein Ladder, ThermoFisher, Illkirch, France).

The global analysis was performed using the method described in [27]. The variation in the total amount of protein analyzed for each individual due to individual variation in the venom gland size and/or total venom protein contents was normalized during both gel staining and image analysis steps as described below. Briefly, the gels were silver stained [42] and photographed several times during the protein revelation step (digital camera EOS-5D-MkII, Canon, Tokyo, Japan) to ensure that all lanes have at least one picture where all shared bands are revealed while not being too saturated (for more details see [13]). These high-resolution pictures (5626 × 3745 pixels; 16 bits; TIFF file) were then analyzed with the Phoretix-1D software (version 11.5, TotalLab, Newcastle-Upon-Tyne, UK) to extract the intensity profile of each lane. The resulting intensity profiles were semi-automatically analyzed using R functions, allowing identification of a dataset of “reference bands” of known molecular weight. The remaining variation in the total intensity of the lanes was normalized with the combination of parameters, “background removed” and “quantile normalization”, and the intensity of the reference bands measured with the variable ‘height’ (see [13]). These normalized intensities are the variables that describe the venom composition.

The specific analysis was performed using Western blots and antibodies against previously characterized *L. boulardi* venom proteins. 1D electrophoretic gels were blotted according to Towbin et al. [43] onto a nitrocellulose membrane (120 V, 1 h; Protran BA85, GE Healthcare Life Sciences, Velizy-Villacoublay, France) that was then incubated 1 h in 2% milk in TBS-Tween (20 mM Tris-HCl, pH 7.3, 150 mM NaCl, 0.2% Tween 20), and overnight at 4 °C in a mix of rabbit polyclonal antibodies directed against LbSPN (1: 2000), LbGAP (1: 10,000)or LbGAP2 (1: 2000). LbSPN antibody was raised against a specific peptide common to the LbSPNm and LbSPNy serpins [10], LbGAP [14] and LbGAP2 [44] antibodies against a recombinant protein. After three washes in TBS-Tween, the membrane was incubated 2 hrs with a secondary goat anti-rabbit IgG horseradish peroxidase conjugate (1: 10,000; Sigma, St. Quentin Fallavier, France) in 2% milk in TBS-Tween, washed three times in TBS-Tween, and revealed with a luminescent substrate (Luminata TM Crescendo Western HRP Substrate; Millipore, Burlington, MA, USA). Digital images of the blot acquired with a cooled CCD camera (Andor iKon-M, Abington-on-thames, UK) were used in the following steps.

Data were then recorded as follows for each individual:

(i) For LbSPN, the genotype at the *lbspn* locus was determined by the presence/absence of the codominant LbSPNy (54 kDa) and LbSPNm (45 kDa) bands.

(ii) For LbGAP, a strong signal is observed in the venom of ISm or (ISm × ISy) F_1_ individuals but not in ISy females. The presence/absence of this dominant band was used to distinguish *lbgap* homozygotes or *lbgap/lbgap_y_* heterozygotes from *lbgap_y_* homozygotes.

(iii) For LbGAP2, the variation in protein quantity is more continuous. We estimated the normalized LbGAP2 quantity as the ratio between the signal intensity on Western blot and a proxy of the amount of venom obtained from the dissection. This proxy consists in the median of the intensities of the reference bands in the corresponding lane on silver stained gels without normalization.

### 4.4. Statistical Analysis for the Global Analysis of Venom Composition

#### 4.4.1. Evolution of the Global Venom Composition

To evaluate whether the composition of the venom evolve differently on the two hosts, we performed a MANOVA that tested the interaction between the generation (F_2_, F_6_ or F_10_; as a continuous variable) and the host genotype, based on the intensity of the reference venom protein bands. Since band intensities were not normally distributed, we used the permutational MANOVA implemented in the “vegan” R package (function adonis2; [45]) with 5000 permutations nested within replicates. This made it possible to evaluate how the Euclidean distances between the venom compositions of individuals are explained by the marginal effect of (i) the host genotype (resistant vs. susceptible; or more accurately the host strain), (ii) the generation (continuous variable), (iii) the interaction between the host genotype and the generation, and (iv) the 16 experimental populations (8 replicates × 2 host strains) to account for the effect of genetic drift. Since the interaction between the host and the generation could either result from a differential evolution of the venom composition on each host or from its evolution on one host only, the MANOVA was also performed for the (R) and (S) hosts separately. In these additional MANOVA, the marginal effects tested were the generation and the replicate. To characterize the global evolution of the venom composition according to the host, we performed a linear discriminant analysis (LDA) with the six groups of individuals: the three analyzed generations (F_2_, F_6_ and F_10_) on the two host strains (R) and (S). For this, we used the “ade4” R package [45] with the individual venom compositions as continuous variables and the combination of host resistance and generation as a factor. Since LDA does not account for the variation between replicates, they were centered before the analysis and significance was assessed using 5000 permutations nested within replicates. The biological meaning of the LDA axes was determined based on the position of the six groups on these axes.

#### 4.4.2. Evolution of Venom Protein Bands

A non-parametric Spearman rank correlation test was performed to describe the evolution of the intensity of each band by testing the correlation between the band intensity and the two first discriminant axes. We then used a linear regression to evaluate the part of the variation of the first axis that was explained by the second axis. Finally, Spearman rank correlation tests were used to determine which bands were correlated to these discriminant axes and identify the ones whose intensity changed on the susceptible or resistant strain. Correlation significance levels were corrected using the Bonferroni procedure with the p.adjust R function. We then used a linear regression to evaluate the part of the variation in the correlations to the first axis that was explained by the correlations to the second axis.

In order to disentangle protein bands that were directly selected from those that were indirectly selected because of their correlation with other bands (partial overlapping of the bands on the gel or linkage disequilibrium), we used a combination of clustering and partial regression analyses. In a first step, a UPGMA clustering analysis (“hclust” R package) was performed using “one minus the absolute value of the correlation between the bands” as the metric distance. Then, a conservative threshold correlation of 0.4 was used to construct band clusters for which one or more directly selected bands might have led to the indirect selection of other bands. For each of these clusters, we determined with partial correlation analyses based on linear regressions if the residual variation in the intensity of each band independent from the other bands of the cluster was still correlated to the discriminant axis. Correlation significance levels were corrected as described just above.

To determine whether most of the protein bands selected on a given host corresponded to ISm or ISy bands, each band also received a value ranging from 0 to 1, in relation to a higher intensity in ISy or ISm, respectively. This value is the intensity of the band in the venom ISm divided by the sum of its intensities in the venoms ISm and ISy.

### 4.5. Statistical Analysis for the Specific Study of Venom Evolution

The three variables that describe LbSPN, LbGAP and LbGAP2 are different in nature (categorical for LbSPN with two different alleles, presence/absence for LbGAP, continuous for LbGAP2 with the corrected intensity) and the analyses to determine whether the amounts of these proteins had evolved differentially on both hosts were different.

In our experiment, the variation of LbSPN and LbGAP has a simple genetic determinism. LbSPN is a codominant marker with two alleles (*lbspn_m_* and *lbspn_y_*) encoding proteins that differ in their molecular weight. LbGAP is a dominant marker with two alleles (*lbgap* and *lbgap_y_*). The simulation approach used to assess whether LbSPNm/LbSPNy and LbGAP/ LbGAPy were selected or counter-selected is described in Appendix A. In short, we computed the expected frequencies of the *lbspn_y_* allele (and therefore of *lbspn_m_*) and of the LbGAP phenotype (sum of the frequencies of the *lbgap/lbgap* and *lbgap/lbgap_y_* genotypes) for the F_6_ and F_10_ generations assuming neutrality and panmixia. They were then compared to the frequencies in the eight replicates for each host strain. The observed difference was evaluated on the set of replicates using a summary statistic whose null distribution was built with the simuPOP simulation software [46]. This software was used to simulate 20,000 times the evolution of a bi-allelic neutral loci evolving in eight haplodiploid populations of the same initial state and size as those in the experiment. The *p*-values were obtained by doubling the proportion of simulated summary statistics that were more extreme than the summary statistic observed (unilateral tests; see Appendix A for more details).

For LbGAP2 (continuous variation), we used a linear mixed model (LMM) to explain the variation of intensity with the host strain, the generation and their interaction as fixed effects, and the experimental populations nested within the replicates as a random effect. The model was fitted using the “nlme” R package [47] on the box cox-transformed (λ = 0.23) corrected intensity of LbGAP2 to achieve normality of residuals (see above for the description of the correction of the intensity of LbGAP2). Then, the model was analyzed with a Tukey test (“multcomp” R package; [48]) to identify which modality differs from the other.

## Figures and Tables

**Figure 1 toxins-11-00629-f001:**
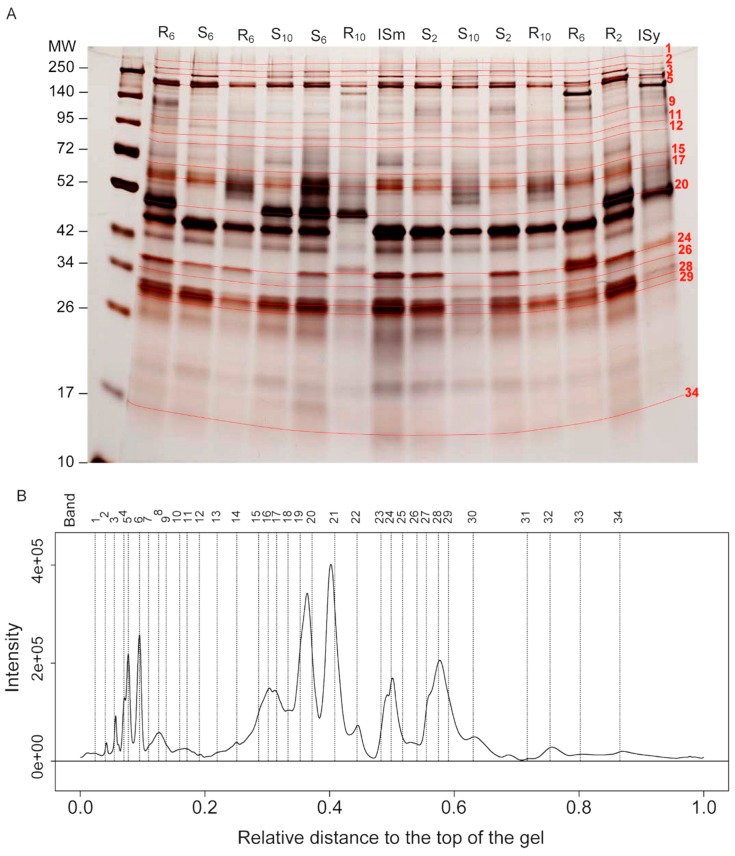
Global analysis of the protein composition of the venom. (**A**) example of venom profiles on silver-stained 1D SDS-PAGE gels. The R and S lanes contain the venom (half of the reservoir) of a single female wasp. They correspond to individuals reared on the resistant (R) or susceptible (S) host strain, respectively. Subscript numbers indicate the generation to which the female belongs. ISm and ISy lanes contain the equivalent of half a reservoir but from a pool of ten gathered reservoirs of ISm and ISy individuals, respectively, used as controls. Red lines (with numbers) correspond to the reference bands identified as selected on at least one host strain. MW: molecular weight in kDa. (**B**) Simplified mean intensity profile obtained by averaging the intensities of each band over all individual profiles using Phoretix 1D and R functions. Vertical dashed lines correspond to the position of the bands on the gel. Relative distance: distance from the top of the gel relative to the height of the gel. Intensity: intensity of the bands in arbitrary units. The profile was simplified for illustrative purposes. A more complete profile is presented in Appendix A. For more details on the procedure to handle overlapping bands, see [27].

**Figure 2 toxins-11-00629-f002:**
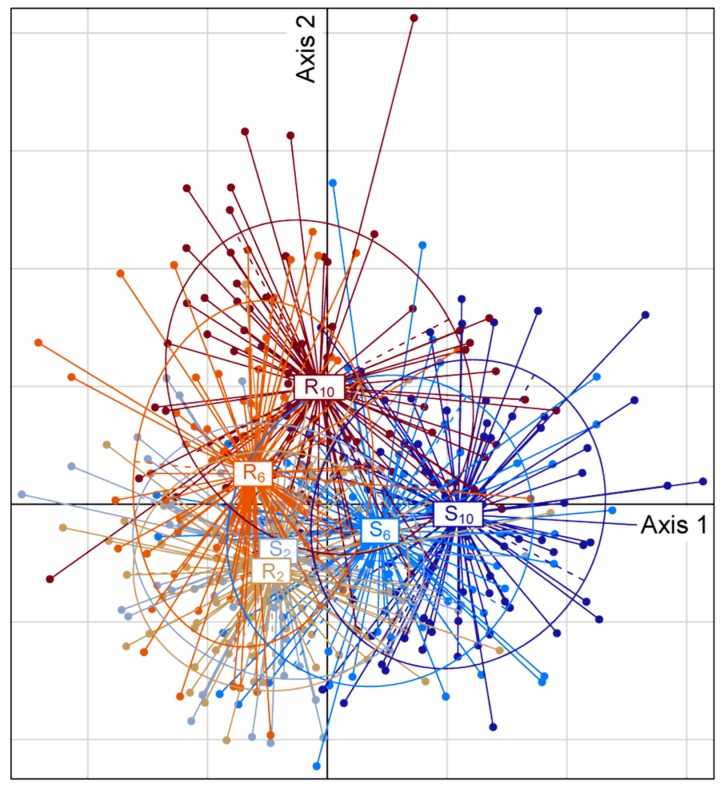
Position of the individuals on the discriminant analysis. Individuals (shown as dots) are grouped and colored according to the host strain (R and S for the resistant and susceptible strain, respectively) and the generation to which the female belong (numbers in subscript).

**Figure 3 toxins-11-00629-f003:**
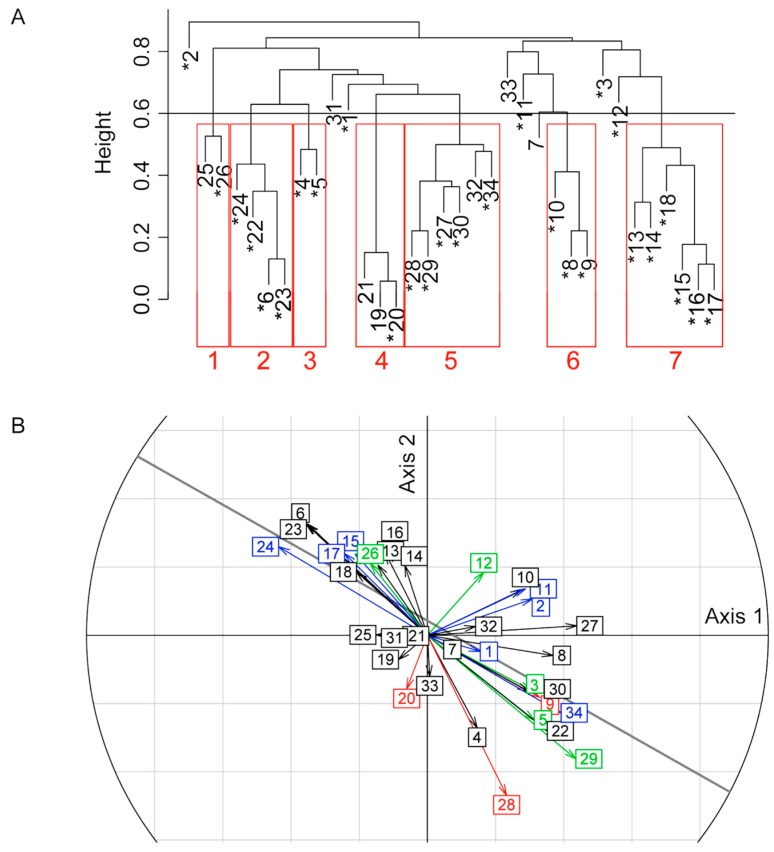
Protein bands correlations with each other and with the discriminant axes. (**A**) clustering analysis. Each numbered leaf of the dendrogram corresponds to a venom protein band. Bands marked with an asterisk correlated with at least one of the first two discriminant axes before the partial correlation analysis. Height represents the independence between bands intensity, calculated as “1 - (absolute value of the correlation between bands intensity)”. The horizontal line at 0.6 represents the 0.4 correlation threshold used to build the seven band clusters (in red) for the partial correlation analysis; (**B**) correlation circle indicating the correlation of bands to the discriminant axes. The numbers correspond to the protein bands. The colors indicate the significance of correlations in the partial correlation analysis: green: correlation to the two axes (bands 3, 5, 12, 26, 29); blue: correlation to the horizontal axis (bands 1, 2, 11, 15, 17, 24, 34); red: correlation to the vertical axis (bands 9, 20, 28); black: no correlation. The oblique line indicates the linear regression between the correlations to axes 1 and 2 (Spearman correlation coefficient of −0.72; *p* < 0.001).

**Figure 4 toxins-11-00629-f004:**
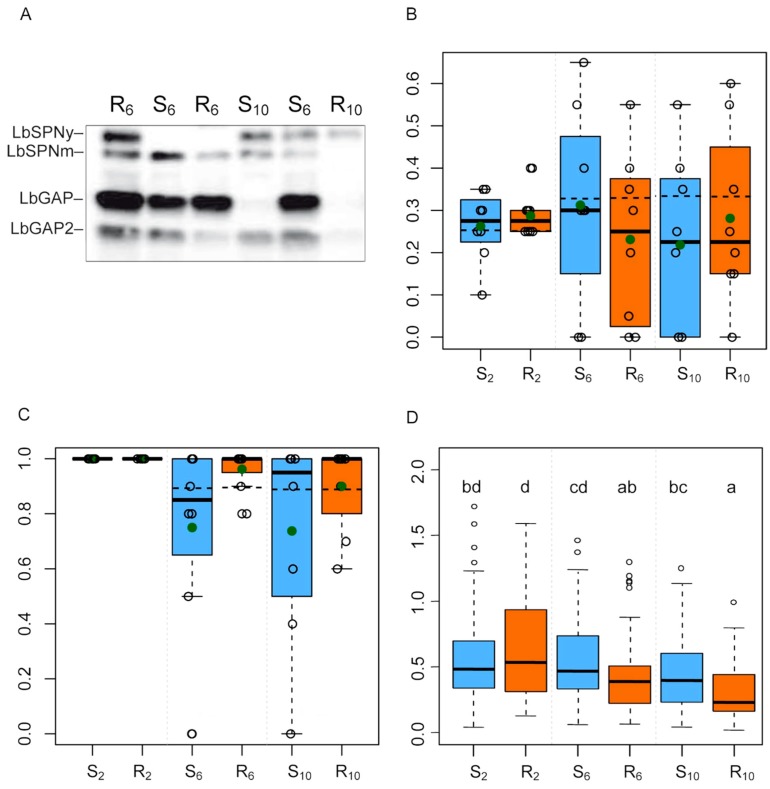
Specific analysis of the evolution of the LbSPN, LbGAP and LbGAP2 proteins. (R) and (S) letters indicate the resistant and susceptible host strains, respectively, and subscript numbers the generation. (**A**) example of Western blot analysis of LbSPN, LbGAP and LbGAP2 proteins for six individuals picked at random. The Western blot was provided for illustrative purposes only and is not representative of overall results. (**B**,**C**) expected and observed frequencies of the *lbspn_y_* (**B**) and *lbgap* (**C**) alleles. The filled green circles indicate the average of the observed frequencies of individuals with an LbGAP phenotype in the experimental populations. The horizontal dashed lines indicate the expected frequencies under neutrality (H0) and panmixia; (**D**) corrected quantity of LbGAP2 measured as the ratio between the signal intensity in Western blot and the median of the intensities of the reference bands in the corresponding lane (proxy of the amount of venom). The same letters indicate no significant difference between groups (LMM, Tukey post hoc test).

**Table 1 toxins-11-00629-t001:** Permutational MANOVA for venom evolution on the host strains and separately on each host train (R) and (S). For each effect considered in the model (generation, host strain (R) vs. (S), and the interaction of the two), the following information is provided. Df: degrees of freedom. Sums of Sqs: sum of squares. F: F statistics. R²: partial R-squared. Pr(>F): *p*-value based on 5000 constrained permutations within replicates.

Model	Variance Partition	Df	Sums of Sqs	F	R²	Pr (>F)
Evolution on the two host strains	Generation	1	3 × 10^9^	3.47	0.01	0.02 *
	Host strain	1	9 × 10^9^	10.36	0.02	2 × 10^−4^ ***
	Generation × Host	1	5 × 10^9^	6.19	0.01	1 × 10^−3^ **
	Population	14	8 × 10^10^	6.87	0.17	2 × 10^−4^ ***
	Residuals	456	4 × 10^11^		0.80	
	Total	473 ^†^	4 × 10^11^		1.00	
Evolution on the (R) host strain	Generation	1	4 × 10^9^	4.28	0.01	1 × 10^−3^ **
	Population	7	3 × 10^10^	5.28	0.14	2 × 10^−4^ ***
	Residuals	231	2 × 10^11^		0.85	
	Total	239 ^†^	2 × 10^11^		1.00	
Evolution on the (S) host strain	Generation	1	4 × 10^9^	5.46	0.02	2 × 10^−3^ **
	Population	7	5 × 10^10^	8.57	0.21	2 × 10^−4^ ***
	Residuals	225	2 × 10^11^		0.77	
	Total	233 ^†^	2 × 10^11^		1.00	

^†^ The degrees of freedom vary between the two permutational MANOVA since only four individuals instead of 10 were available for venom analysis in one F_2_ replicate on the (S) host.

**Table 2 toxins-11-00629-t002:** Summary of bands evolution in response to the host strain. (i) first columns: Sign (+ and -) and significance (* < 0.05; ** < 0.01; *** < 0.001) of the bands intensity correlation with the two discriminant axes. Among clustered bands, those numbered in bold are significantly correlated with at least one of the discriminant axes (before and after the partial correlation analysis) and thus considered as selected. Non-significant correlations are in italics. (ii) Last columns: direction of evolution of the bands intensity (up arrow: positive selection; down arrow: negative selection) according to their origin that was estimated by dividing the band intensity in ISm venom by the sum of its intensity in ISm and ISy venoms. ISy = 0; ISm = 1. Bands not correlated to an axis are not shown. For complete data, see Appendix A.

Band Number	Cluster Number	Correlation with Axis 1 (Evolution on S Strain)				Correlation with Axis 2 (Evolution on R Strain)				Band Origin	Band Evolution According to the Origin			
		Before Partial Correlations		After Partial Correlations		Before Partial Correlations		After Partial Correlations			S Strain		R Strain	
											ISm	ISy	ISm	ISy
1	None	+	*			-	n.s.			0.84	↗			
2	None	+	***			+	n.s.			0.70	↗			
3	None	+	***			-	*			0.86	↗		↘	
11	None	+	***			+	n.s.			0.24		↗		
12	None	+	*			+	**			0.38		↗		↗
26	1	-	*	-	*	+	***	+	***	0.61	↘		↗	
6	2	-	***	-	n.s.	+	***	+	n.s.	0.12				
22	2	+	***	+	n.s.	-	***	-	n.s.	0.88				
23	2	-	***	+	n.s.	+	***	+	n.s.	0.03				
24	2	-	***	-	***	+	***	+	n.s.	0.62	↘			
4	3	+	n.s.	+	n.s.	-	***	-	n.s.	0.46				
5	3	+	***	+	***	-	***	-	*	0.60	↗		↘	
20	4	-	n.s.	-	n.s.	-	*	-	*	0.06				↘
27	5	+	***	+	n.s.	+	n.s.	+	***	0.65				
28	5	+	***	-	**	-	***	-	***	0.79			↘	
29	5	+	***	+	***	-	***	-	***	0.82	↗		↘	
30	5	+	***	-	n.s.	-	*	+	n.s.	0.85				
34	5	+	***	+	**	-	***	-	n.s.	0.93	↗			
8	6	+	***	+	n.s.	-	n.s.	-	n.s.	0.10				
9	6	+	***	+	n.s.	-	**	-	***	0.43				↘
10	6	+	***	+	n.s.	+	n.s.	+	n.s.	0.07				
13	7	-	n.s.	-	n.s.	+	***	-	n.s.	0.77				
14	7	-	n.s.	-	n.s.	+	***	+	n.s.	0.69				
15	7	-	***	-	**	+	***	+	n.s.	0.48		↘		
16	7	-	n.s.	-	n.s.	+	***	+	n.s.	0.43				
17	7	-	***	-	***	+	***	-	n.s.	0.24		↘		
18	7	-	***	+	n.s.	+	**	+	n.s.	0.07				

**Table 3 toxins-11-00629-t003:** Correspondence between evolving bands and their putative protein content. Correspondence based on the comparison with data from [10]. Only proteins for which at least 10 peptide matches were found in Mascot searches were considered as abundant and listed. The number of proteins in the band, their predicted function, and the number of peptides matches for each unisequence are provided. Data on the band origin and direction of evolution are from Table 2. Up and down arrows indicate a selection or a counter-selection on the corresponding host strain, respectively.

Reference Band	Number of Proteins in the Band	Putative Function	Number of Peptides Matches	Band Origin	Evolution on the R Strain	Evolution on the S Strain
3	1	Unknown	12	ISm	↘	↗
5	6	Unknown	49	ISm	↘	↗
		Unknown	46			
		Unknown	34			
		Sushi/SCR/CCP domain containing protein	18			
		Unknown	14			
		Unknown	11			
11	2	Sushi/SCR/CCP domain containing protein	10	ISy		↗
		Unknown	10			
15	3	Unknown ^a^	39	ISy		↘
		Unknown ^b^	36			
		Unknown	28			
17	2	Unknown ^a^	58	ISy		↘
		Unknown ^b^	36			
20	1	Serpin (LbSPNy)	81	ISy	↘	
24	5	RhoGAP (LbGAP)	52	ISm		↘
		Unknown	21			
		Serpin (LbSPNm)	17			
		Unknown	12			
		Unknown	11			
26	1	RhoGAP (LbGAPy4)	24	ISm	↗	↘
28	3	RhoGAP (LbGAP2)	43	ISm	↘	
		RhoGAP (LbGAP1)	23			
		Serpine (LbSPNm)	15			
29	4	RhoGAP (LbGAP2)	68	ISm	↘	↗
		Unknown	21			
		RhoGAP (LbGAPy2)	19			
		Unknown	11			

^a,b^ Same protein with unknown function found in two different bands.

## Data Availability

The data sets supporting the results will be made available in the figShare repository under the DOI: 10.6084/m9.figshare.10060691.

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
