# Peer review of "Rapid and Differential Evolution of the Venom Composition of a Parasitoid Wasp Depending on the Host Strain"

_toxins, 2019, doi:10.3390/toxins11110629_

Round 1

Reviewer 1 Report

This study tries to show the short-term evolution by observing the change of total protein band profiles on SDS-PAGE. Simple comparison of shifted bands or the intensity change in each unknown protein band makes inaccurate information. As every person knows, we are hard to determine that the protein band electrophoresed at same position was similar protein on SDS-PAGE. Of course, like authors presented, it will be understood for readers only when authors focused on the specific proteins like RhoGAP or SNP.  Further it is well known that in host-parasitoid interaction, the results caused by selection-pressure appears soon, because the poor development of parasitoid larvae in unsuitable host physiological condition causes poor next generation. Authors seem to have already determined some protein bands on SDS-PAGE band profile. I would like to recommend authors to re-analyze the band profile by focusing on the already clarified bands like LbGAP, LbSNP which are related with regulation of host defense reaction. However, it is one of very careful points that each peak, especially near-peaks overlapped when authors use the densitometoric pattern analysis, for example in this paper the band#10-11 or the bands#27-28-29 overlapped, it should be problem how to measure the relative distance and intensity of each band and authors will be required to explain it with technical procedures precisely. The band#10-11, bands#27-28-29should be described how to discriminate the band#11, 20, 24 26-28,29, respectively. In analysis of Fig.3A, it seems to show the close relation among near band profile like 25 and 26 or 19, 20, 21 and so on.

Further, the results that the protein combined from 10 venom reservoirs was electrophoresed per a lane seem to ignore the individual difference, which is important when authors want to assess the accuracy of the global profile as you understand.

Terminological problem is “evolution”, evolve or evolution used in this paper makes a little misunderstanding, I think. “change”? If the term “evolution” means that “short-term change by strong selection pressure”, recommend to use more suitable word.

I cannot understand Fig 4; I recognized the different band profiles in same R6-R6 and S6-S6 lanes in panel A, LbGAP was disappeared in both R10 and S10 strains. In C and D panels their results seem not to be reflected. Both of LbGAP in panel C and LbGAP2 in panel D based on the result of panel A seems to disappear. It seems that Fig 4 cannot support readers to understand the change of specific proteins like LbSNP and LbGAP through generation easily. More precise information will be required.

Author Response

We are grateful to the reviewer for his comments that helped to improve the manuscript by highlighting parts that deserved more explanation for a good understanding of the method and the analysis. Please see below the response to the different comments.

Comment 1: Simple comparison of shifted bands or the intensity change in each unknown protein band makes inaccurate information. As every person knows, we are hard to determine that the protein band electrophoresed at same position was similar protein on SDS-PAGE.

We agree that the protein composition of the bands found at a given position in the SDS-PAGE may not be exactly the same between individuals, as we already mentioned in the manuscript in lines 206-210. We have clarified this further in the revised version. However, the statistical analysis method on which this study relies (developed and published in Mathé-Hubert et al. (2015; https://doi.org/10.1111/1755-0998.12389)) can handle the variation in intensity that may result from this variation in protein composition. In addition, the congruence we observed between the global semi-automatic analysis and the Western blot analysis for well-characterized venom proteins made it possible to test and confirm the accuracy of the first approach.

 Comment 2: I would like to recommend authors to re-analyse the band profile by focusing on the already clarified bands like LbGAP, LbSNP which are related with regulation of host defence reaction.

This is why we performed a Western blot analysis for these well-characterized venom proteins, a more accurate analysis than the global semi-automated method through qualitative or quantitative antibody evaluation. The congruence observed between the two analyses was used as an additional source of validation for the global analysis, the accuracy of which had already been evaluated in our previous work (Mathé-Hubert et al. 2015; https://doi.org/10.1111/1755-0998.12389). It should be noted however that our study aimed to identify new proteins whose quantitative change during the experimental evolution (visualized by the intensity of the band containing them) suggested their possible involvement in the success of the parasitoid.

 Comment 3: However, it is one of very careful points that each peak, especially near-peaks overlapped when authors use the densitometric pattern analysis, for example in this paper the band#10-11 or the bands#27-28-29 overlapped, it should be problem how to measure the relative distance and intensity of each band and authors will be required to explain it with technical procedures precisely.

The band#10-11, bands#27-28-29 should be described how to discriminate the band#11, 20, 24 26-28,29, respectively. In analysis of Fig.3A, it seems to show the close relation among near band profile like 25 and 26 or 19, 20, 21 and so on.

The mean intensity profile shown in Figure 1B was simplified for illustrative purpose. It was obtained by averaging all the profiles using the method thoroughly described in our previous work (Mathé-Hubert et al. (2015; https://doi.org/10.1111/1755-0998.12389)). This method treats  the overlapping bands using the second derivative for which the results are not shown in figure 1B. Since the average of the profiles induces a smoothing, one could have the impression that the overlap between the bands is greater than it really is. For example, bands #10 and 11 seem to be indistinguishable in Figure 1B but are clearly separated on the gel picture (Figure 1A). The same can be seen for bands #27-28-29. We have added in the supplementary material a more complete profile including the results of using the second derivative (Supplementary Figure S3). Overlap between the bands nevertheless occurred and the intensity of the strong bands affected the intensity of the small overlapping bands. This has induced artifactual correlations between the overlapping bands and we accounted for this using partial correlations as shown in Figure 3A.

Comment 4: Further, the results that the protein combined from 10 venom reservoirs was electrophoresed per a lane seem to ignore the individual difference, which is important when authors want to assess the accuracy of the global profile as you understand.

The venom reservoirs of individuals from the experimental evolution were indeed analysed separately with a single individual per lane (half a reservoir). The merging of the venom of 10 individuals was only done for the control lanes corresponding to the ISm and ISy lines. The purpose of these control lanes was to obtain a reference profile on each gel to help analysing the gel pictures. We have clarified this point in the manuscript (lines 104-105; Figure 1 legend lines 118-122; Mat and Meth lines 483-489).

Comment 5: Terminological problem is “evolution”, evolve or evolution used in this paper makes a little misunderstanding, I think. “change”? If the term “evolution” means that “short-term change by strong selection pressure”, recommend using more suitable word.

A rapid change induced by a strong selection pressure is an evolution and we prefer to keep this term which seems to us the most relevant for readers interested by experimental evolution approaches. If the reviewer/editor really want a change, we can propose "adaptation" (used in some places in the revised manuscript) which is suitable when referring to evolution induced by selection and not drift. However, not all bands change their intensity due to selection in our analysis (drift may be responsible for some), making the term "adaptation" irrelevant for some of the data.

Comment 6: I cannot understand Fig 4; I recognized the different band profiles in same R6-R6 and S6-S6 lanes in panel A, LbGAP was disappeared in both R10 and S10 strains. In C and D panels their results seem not to be reflected. Both of LbGAP in panel C and LbGAP2 in panel D based on the result of panel A seems to disappear. It seems that Fig 4 cannot support readers to understand the change of specific proteins like LbSNP and LbGAP through generation easily. More precise information will be required.

Figure 4A was only provided for illustrative purpose. Each of the six lanes shown corresponds to an individual picked at random. These individuals are therefore not representative of overall results. We have clarified this point in the legend of the figure (lines 271-272).

The reviewer also stated that: The manuscript does need some careful editing for Grammar and word choice. We have taken this comment into account by correcting for the grammar and choice of words in different parts of the manuscript.

Reviewer 2 Report

Overall, I really enjoyed reading this manuscript. The findings presented and model system were both really interesting and intriguing and I especially liked the experimental evolution approach. In fact, this was one of the most exciting manuscripts I have read recently and the authors should be commended for their excellent work. The manuscript does need some careful editing for Grammar and word choice, but I will leave that up to the editorial team. I wonder if in the future the authors could use a mapping approach to understand the the differences in venom composition among the different groups.

Author Response

We are thankful to the reviewer for its positive comments on the work performed and its interest in the results.

We will carefully consider its suggestion of using a mapping approach to understand differences in venom composition between different groups, in our future work.

Both reviewers also stated that: The manuscript does need some careful editing for Grammar and word choice. We have taken this comment into account by correcting for the grammar and choice of words in different parts of the manuscript.

Round 2

Reviewer 1 Report

This paper became understandable easily, but some sentences in Line 62-67 are not clear yet. More kind explanation will be required for readers. I think these sentences seem not to be important for this paper. Try to simplify or modify this to be more understandable sentence to readers.

Author Response

Comments from the Reviewer

"This paper became understandable easily, but some sentences in Line 62-67 are not clear yet. More kind explanation will be required for readers. I think these sentences seem not to be important for this paper. Try to simplify or modify this to be more understandable sentence to readers."

We have used well-characterized resistant and susceptible strains of Drosophila in the experiments and we think it is important for readers to have at least some information on how these strains were obtained, and how the resistance is determined. Besides, Rlb was the first resistance gene to parasitoids identified. References are also important in this context for those that may want more details.
The genetics of Drosophila is a particular area and some information in these lines was very specific. It was consequently suppressed (see manuscript review). We have also rewritten lines 62-67 by simplifying and clarifying further to facilitate readers' understanding.

Thanks for suggesting these modifications.